# Simultaneous Bilateral Video–Endoscopic Inguinal Lymphadenectomy for Penile Carcinoma: Surgical Setting, Feasibility, Safety, and Preliminary Oncological Outcomes

**DOI:** 10.3390/jcm12237272

**Published:** 2023-11-23

**Authors:** Josep M. Gaya, Giuseppe Basile, Pavel Gavrilov, Andrea Gallioli, Angelo Territo, Jorge Robalino, Pedro Hernandez, Raul Sanchez-Molina, Alejandra Bravo, Ferran Algaba, Jordi Huguet, Francesco Sanguedolce, Joan Palou, Antonio Rosales, Alberto Breda

**Affiliations:** 1Department of Urology, Fundació Puigvert, Autonomous University of Barcelona, 08025 Barcelona, Spain; jmgaya@fundacio-puigvert.es (J.M.G.); basile.giuseppe@hsr.it (G.B.); pgavrilov@fundacio-puigvert.es (P.G.); agallioli@fundacio-puigvert.es (A.G.); aterrito@fundacio-puigvert.es (A.T.); jrobalino@fundacio-puigvert.es (J.R.); phernandez@fundacio-puigvert.es (P.H.); rsanchez@fundacio-puigvert.es (R.S.-M.); abravo@fundacio-puigvert.es (A.B.); jhuguet@fundacio-puigvert.es (J.H.); jpalou@fundacio-puigvert.es (J.P.); arosales@fundacio-puigvert.es (A.R.); abreda@fundacio-puigvert.es (A.B.); 2Department of Urology, Vita-Salute San Raffaele University, 08025 Milan, Italy; 3Department of Pathology, Fundació Puigvert, Autonomous University of Barcelona, 08025 Barcelona, Spain; falgaba@fundacio-puigvert.es; 4Department of Medicine, Surgery and Pharmacy, Sassari University, 07100 Sassari, Italy

**Keywords:** penile carcinoma, video–endoscopic inguinal lymphadenectomy, inguinal lymph node dissection, minimally invasive surgery, invasive inguinal staging

## Abstract

Introduction: Inguinal lymph node dissection (ILND) plays an important role for both staging and treatment purposes in patients diagnosed with penile carcinoma (PeCa). Video–endoscopic inguinal lymphadenectomy (VEIL) has been introduced to reduce complications, and in those patients elected for bilateral ILND, a simultaneous bilateral VEIL (sB-VEIL) has also been proposed. This study aimed to investigate the feasibility, safety, and preliminary oncological outcomes of sB-VEIL compared to consecutive bilateral VEIL (cB-VEIL). Material and methods: Clinical N0-2 patients diagnosed with PeCa and treated with cB-VEIL and sB-VEIL between 2015 and 2023 at our institution were included. Modified ILND was performed in cN0 patients, while cN+ patients underwent a radical approach. Intra- and postoperative complications, operative time, time of drainage maintenance, length of hospital stay and readmission within 90 days, as well as lymph node yield, were compared between the two groups. Results: Overall, 30 patients were submitted to B-VEIL. Of these, 20 and 10 patients underwent cB-VEIL and sB-VEIL, respectively. Overall, 16 (80%) and 7 (70%) patients were submitted to radical ILND due to cN1-2 disease in the cB-VEIL and sB-VEIL groups, respectively. No statistically significant difference emerged in terms of median nodal yield (13.5 vs. 14, *p* = 0.7) and median positive LNs (*p* = 0.9). sD-VEIL was associated with a shorter operative time (170 vs. 240 min, *p* < 0.01). No statistically significant difference emerged in terms of intraoperative estimated blood loss, length of hospital stay, time to drainage tube removal, major complications, and hospital readmission in the cB-VEIL and sB-VEIL groups, respectively (all *p* > 0.05). Conclusions: Simultaneous bilateral VEIL is a feasible and safe technique in patients with PeCA, showing similar oncological results and shorter operative time compared to a consecutive bilateral approach. Patients with higher preoperative comorbidity burden or anesthesiological risk are those who may benefit the most from this technique.

## 1. Introduction

Penile carcinoma (PeCa) is a rare malignant disease characterized by heterogeneous clinical behavior [1]. The earliest site of regional dissemination of PeCa is the inguinal lymph nodes, whose involvement represents the most important prognostic factor for survival [1,2]. Inguinal nodal involvement is found in up to 30% of newly diagnosed cases, and approximately 20–25% of clinical N0 patients may harbor occult metastases [1,3]. Thus, according to uro-oncological guidelines, regional nodal assessment plays a pivotal role for both staging (in intermediate and high-risk cN0 patients) and treatment (cN1-2 patients) purposes [1]. Dynamic sentinel lymph node biopsy (DSNB) is considered the first recommended option in cN0 patients, supported by similar accuracy and less morbidity compared to inguinal lymph node dissection (ILND), although accuracy varies among centers [1,4]. Furthermore, DSNB is not widespread outside referral centers or may not be feasible due to tumor characteristics. Thus, ILND remains the only alternative option. Open ILND (OILND) has been historically considered the gold-standard approach for invasive inguinal node assessment in patients diagnosed with PeCa as a modified template for cN0 and radical for cN+ patients. However, both templates have been associated with a high morbidity burden and long sequelae [1,5]. Since it was first described in 2006, minimally invasive video–endoscopic inguinal lymphadenectomy (VEIL) has led to significant advantages as compared to the open approach by reducing postoperative morbidity while maintaining comparable oncological outcomes [6,7,8,9]. Furthermore, several other refinements have been attempted to improve surgical outcomes through the introduction of a robotic approach [10,11,12]. Similarly, in patients elected for bilateral ILND, a simultaneous bilateral VEIL (sB-VEIL) technique has been proposed to further reduce anesthesiology and surgery-related morbidity compared to a consecutive bilateral procedure [13,14,15]. In this study, we describe the technique and surgical setting of sB-VEIL, investigating its feasibility, safety, and preliminary oncological outcomes compared to consecutive bilateral VEIL (cB-VEIL) in patients diagnosed with PeCa and elected for bilateral ILND both for staging and treatment purposes. 

## 2. Materials and Methods

### 2.1. Study Design and Variables Definition

After Institutional Ethical Board approval (2014/17), we relied on a single institution prospectively maintained database including individuals who underwent VEIL for PeCa between 2015 and 2023. For study purposes, only patients submitted to B-VEIL for cN0-2 disease were included. Before January 2021, all patients elected for bilateral lymph node dissection underwent cB-VEIL, and then sB-VEIL was performed. 

Baseline variables regarding patients’ characteristics included age at surgery, body mass index, smoking status, and history of diabetes. Preoperative anesthesiology patients’ risk profile was assessed using the American Society of Anesthesiologists (ASA) score [16]. All patients underwent resection of the primary PeCa with glansectomy, partial or radical penectomy according to the cT stage [1]. Clinical nodal status was based on clinical examination and imaging. Ultrasound-guided fine needle aspiration biopsy was considered when suspicious LNs were found at clinical examination or imaging. An abdominal contrast-enhanced CT scan was used to complete the staging process and to assess pelvic nodes if necessary. 

A modified or radical ILND template was performed both in cB-VEIL and sB-VEIL cases depending on clinical node staging as recommended by uro-oncological guidelines [1]. All procedures were performed by two surgeons (J.M.G. and P.G.), both with extensive experience in inguinal lymphadenectomy for PeCa and laparoscopic surgery. Clinical information on primary tumor pathological T stage, grade, and presence of lymphovascular invasion, as well as cN stage, were available for all patients. All specimens of primary tumor excision and ILND were examined by the same dedicated genitourinary pathologists’ team. The number of LNs dissected and of positive LNs were also recorded since it has been postulated as a surrogate for adequacy of LN dissection [17,18]. Intra- and postoperative complications, operative time, time of drainage maintenance, as well as the length of hospital stay and readmission within 90 days after surgery were also recorded. Postoperative complications were graded according to the Clavien–Dindo classification. Patients were followed up with clinical visits and cross-sectional imaging according to uro-oncological guidelines and tumor risk profiles. The length of the follow-up was considered the last information on file until the tumor recurrence or patient death date.

### 2.2. Outcome Measurements and Statistical Analysis

Descriptive statistics included frequencies and proportions for categorical variables. Means, medians, and ranges were reported for continuously coded variables. Chi-square, *t*-test, and Kruskal–Wallis tests were performed to examine the statistical significance in proportion, mean, and median differences. Statistical analysis was performed using R studio Inc. (2016) integrated development environment for R software v. 3.5.3, Boston, MA (USA). All tests were two-sided with a level of significance set at *p* < 0.05.

### 2.3. Patient Positioning, Trocar Placement and Surgical Setting

The procedure is carried out under general anesthesia, and antibiotic prophylaxis is administered during anesthesia induction. The patient is placed in a supine position with the arms along the body, the hips abducted at 45°, and the knees flexed at 90° using Allen stirrups (Figure 1A,B). The Scarpa’s triangle is marked along with its landmarks: Inguinal or Poupart ligament, sartorius muscle, and adductor longus muscle. A 3-port configuration is used (Figure 2A,B). The first incision is performed approximately 2 cm distal to the apex of the femoral or Scarpa’s triangle. The working space is then created using blunt finger dissection between the surface of the fascia lata (deeply) and the Camper’s fascia (superficially). Once the surface of the fascia lata is reached, it is separated from the overlying fatty tissue. A 12 mm balloon trocar is then placed through the first incision to inflate CO_2_ at 8–9 mmHg and is used as a camera port. Two other operative 5 mm balloon trocars are placed approximately 5 cm from the first incision and 2 cm outside the lateral and medial margin of the femoral triangle, respectively. In sB-VEIL, two assistant surgeons stand on the patient leg’s medial part while principal surgeons are placed laterally. Two independent laparoscopic towers and screens are used and placed contralaterally to the surgical field and the principal surgeon (Figure 3A,B).

### 2.4. Surgical Technique and Postoperative Care

The limits of the ILND are the adductor longus muscle (medially), the sartorius muscle (laterally), the Poupart inguinal ligament (superiorly), the Camper’s fascia (superficially), and the femoral vessels and fascia lata (deeply). The dissection starts from the apex of the femoral triangle by separating the fat tissue between the medial margin of the sartorius and adductor longus muscle until the great saphenous vein is identified. Dissection is performed using monopolar or single-use sealing devices in a caudal-to-cranial fashion. Once the great saphenous vein is identified in the roof of the femoral triangle, close to the medial margin of the adductor longus muscle, it is skeletonized until the saphenous-femoral vein junction is reached. Small perforating vascular branches encountered during the dissection may be spared. Otherwise, they are controlled using 3 mm titanium clips or 5 mm non-absorbable clips for vessel ligation. The surgical field is expanded cranially until the Poupart inguinal ligament is identified superior-laterally. In the modified ILND template, once the inguinal ligament is identified, the dissection of superficial lymph nodes is performed corresponding to the Daseler areas I, IV, and V, while the great saphenous vein and its tributaries are usually spared [19]. At this step, a careful evaluation of the femoral vessels and femoral nerve course should be performed to avoid their injury. The dissected tissue is removed using a laparoscopic Endobag and extracted through the camera port incision. A 12Fr tube drain with vacuum suction is placed through the lateral 5 mm port incision in each leg. The port incisions are closed with 3/0 vicryl rapid stitches. Elastic compression stockings are placed at the end of the surgery and maintained for at least 24 h. Deambulation is initiated at 24 h, and subcutaneous low-weight molecular heparin is used for up to 30 days. The patient is discharged with the inguinal drains in situ and evaluated at outpatient clinical visits for wound inspection and drain removal. Large-spectrum antibiotic prophylaxis is maintained for one week after surgery. The drains are removed when the output is less than 10 mL/day.

## 3. Results

Overall, 30 patients were submitted to B-VEIL. Of these, 20 and 10 patients underwent cB-VEIL and sB-VEIL, respectively. The median [interquartile range (IQR)] follow-up for patients who underwent cB-VEIL and sB-VEIL was 46.8 (IQR 33.4–65.9) months and 13 (IQR 2.6–21.3) months, respectively. There was no statistically significant difference in terms of age, body mass index, clinical lymph node status, lymphovascular invasion, and grade of the primary tumors between cB-VEIL and sB-VEIL patients (all *p* > 0.05). Conversely, patients submitted to sB-VEIL had more frequently lower pT stage of the primary tumor (*p* = 0.03) (Table 1). Overall, 16 (80%) and 7 (70%) patients were submitted to radical ILND due to cN1-2 disease in the cB-VEIL and sB-VEIL groups, respectively. No statistically significant difference emerged in terms of median nodal yield (13.5 vs. 14, *p* = 0.7) and median positive LNs (*p* = 0.9) between cB-VEIL and sB-VEIL. Furthermore, both nodal yield and median positive LNs were comparable when considering the surgical side (all *p* > 0.05).

All surgical procedures were completed successfully without any intraoperative adverse events or interruption of the surgery. sD-VEIL was associated with a shorter operative time (170 vs. 240 min, *p* < 0.01). Conversely, no statistically significant difference emerged in terms of intraoperative estimated blood loss, length of hospital stay, and time to drainage tube removal (all *p* > 0.05). When considering postoperative morbidity, 3/20 (15%) and 1/10 (10%) patients developed postoperative major complications (Clavien–Dindo ≥ IIIa) in the cB-VEIL and sB-VEIL groups, respectively. Furthermore, no statistically significant difference emerged in terms of wound, lymphedema, lymphocele, and deep vein thrombosis complications between the two groups (all *p* > 0.05). Moreover, no patients developed clinically significant acidosis or hypercapnia. Finally, no statistically significant difference emerged in terms of hospital readmission within 90 days between cB-VEIL and sB-VEIL (*p* = 0.9). A comprehensive description of recorded postoperative complications according to treatment group is provided in Table 2. Two and one patients had local recurrence and disease progression in the cB-VEIL group, respectively, with two tumor-related deaths within the follow-up period. Conversely, only one patient experienced local recurrence in the sB-VEIL group, with no tumor-related death.

## 4. Discussion

The objective of ILND for individuals with PeCa is to obtain precise local staging, provide guidance for adjuvant treatment choices, and enhance overall survival rates. ILND is the gold-standard treatment for inguinal node-positive patients, and its staging role has been refined in cN0 disease after the introduction of DSNB [1,9,12]. Nevertheless, ILND continues to play a role in this setting since false negative cases after a DSNB are still an issue while its availability is scarce [4]. Although OILND has been demonstrated to be an effective procedure, the significant associated morbidity has been a considerable burden for both surgeons and patients. In this regard, the minimally invasive approach as laparoscopic (VEIL) or robotic (RA-VEIL) has been proven to reduce postoperative morbidity while maintaining the oncological efficacy [7,8,9]. Furthermore, a simultaneous approach has been proposed for both staging and treatment purposes in patients elected for bilateral invasive inguinal node assessment to further decrease surgery-related complications. In this study, we provided insight into the surgical setting of sB-VEIL while providing evidence regarding the feasibility, safety, and preliminary oncological outcomes.

The first description of the procedure was published by Pompeo A. et al. and Herrel L. et al., who demonstrated that bilateral endoscopic lymphadenectomy can be performed simultaneously and could significantly decrease both anesthesiology morbidity and operative time [14,15]. Recently, Ma S. and colleagues reported the first large series of patients treated with sB-VEIL and compared their outcomes with those of patients submitted to OILND and cB-VEIL [13]. Preliminary findings revealed that sB-VEIL provides adequate oncological control, lowering the morbidity of the open approach while being more efficient in terms of operative time as compared to cB-VEIL. In this study, we confirmed that sB-VEIL is a feasible technique that does not increase the postoperative morbidity of the surgery while being more time-saving as compared to cB-VEIL. Nevertheless, despite our results being in line with those reported by recent systematic reviews [11,12,20], a non-negligible portion of patients are at risk of developing postoperative complications after VEIL, with lymphocele and symptomatic lymphedema being the most frequent. Thus, although several innovations have been pursued to reduce surgery-related morbidity, patients treated with ILND have an intrinsic risk of developing complications, regardless of the approach. In this regard, DSNB certainly represents a valuable tool, at least in cN0 patients, to decrease morbidity; however, its learning curve, moderate specificity, reduced availability, and non-standardized technique still limit its usefulness [4,21,22].

Furthermore, we confirmed the safety of sB-VEIL in terms of specific complications such as hypercapnia and acidosis. Although a clear pathophysiological pattern has not been already demonstrated, several studies have postulated and reported an increased risk of acidosis and hypercapnia in extraperitoneal surgeries due to increased carbon dioxide absorption as compared to intraperitoneal ones [23,24]. Thus, hypothetically, a simultaneous extraperitoneal surgery may further increase that risk. In our study, patients submitted to sB-VEIL did not show a higher risk of acidosis and hypercapnia, as also reported by Ma S. et al. [13]. In this regard, both the maintenance of a constant low pressure of 8–9 mmHg and the reduced overall operative time represent fundamental elements to reduce the risk of such types of complications.

Finally, sB-VEIL also appeared effective in terms of oncological outcomes. The number of LNs dissected has been proposed as a surrogate of ILND adequacy, but the definition of threshold is not uniform in the literature [1]. Although some studies showed that a number ≥ 7 of unilateral nodes harvested reflects a reliable oncological procedure, others considered a yield of 15 LNs since it has been associated with improved overall survival [17,18]. In this regard, a recent meta-analysis, including 8 studies comparing OILND with different minimally invasive approaches for radical ILND, considered 12 LNs as a novel threshold of adequacy [11]. In our study, considering both modified and radical procedures, the median number of LNs dissected was 7 per each side, with no statistically significant difference between the two groups of patients. Thus, although survival results and longer follow-up data are needed to correctly evaluate the oncological efficacy of sB-VEIL compared to the open approach, preliminary findings are intriguing and in line with those of previous VEIL series.

According to our findings, sB-VEIL is a feasible procedure with a comparable safety profile to cB-VEIL, with the main strength of reduced operative time and potential cost-saving. Although the simultaneous approach requires the use of several surgical instruments twice, most of them are reusable disposable (monopolar scissors, bipolar forceps, cameras) except for single-use sealing devices. Thus, it could reduce surgery-related costs due to the lower operative time, and in patients elected for bilateral ILND, it reduced anesthesia-related costs as well as hospitalization costs. Nonetheless, additional data is imperative to thoroughly evaluate the cost-effectiveness of this procedure. Furthermore, sB-VEIL surgery may ease the implementation of mentoring programs in laparoscopic surgery since young or inexperienced surgeons could be guided side by side during the surgical steps by experts in the field. In this regard, the surgical setting presented herein allows for an ergonomic procedure in both limbs without any technical disadvantage or issues compared to cB-VEIL.

Finally, although further data are needed to confirm its efficacy in terms of survival, preliminary oncological outcomes are comparable to those of the previous series.

Thus, taking all these findings together, adopting a simultaneous approach should be considered to mitigate the potential for anesthesia-related complications and to decrease surgical-related expenses.

Simultaneous ILND represents an alternative to a consecutive approach, especially in inguinal node-positive patients elected for bilateral ILND with high preoperative anesthesiological risk or in cN0 disease in the absence of non-invasive diagnostic tools such as DSNB.

Our study is not devoid of limitations. First, the retrospective analysis of the study represents an intrinsic limitation. Second, the study was based on a single-center experience with expert laparoscopic surgeons involved. Thus, multicenter studies are needed to demonstrate the feasibility of the technique and the possible advantages of a simultaneous approach in terms of reduced morbidity compared to a consecutive technique, as well as estimate its learning curve in laparoscopic-naïve or inexperienced surgeons and its cost-effectiveness. Third, we included both patients who underwent ILND for node-positive and cN0 disease, which may be associated with different postoperative outcomes due to local tissue invasion/necrosis and extension of dissection. Fourth, the sample size of the study is small, in part because of the rarity of PeCa. Fifth, the relatively short follow-up of patients treated with sD-VEIL.

## 5. Conclusions

Simultaneous bilateral video–endoscopic inguinal lymphadenectomy is a feasible and safe technique in patients with penile carcinoma, showing similar oncological results and shorter operative time compared to a consecutive bilateral technique. Patients with higher preoperative comorbidity burden or anesthesiological risk are those who may benefit the most from this technique.

## Figures and Tables

**Figure 1 jcm-12-07272-f001:**
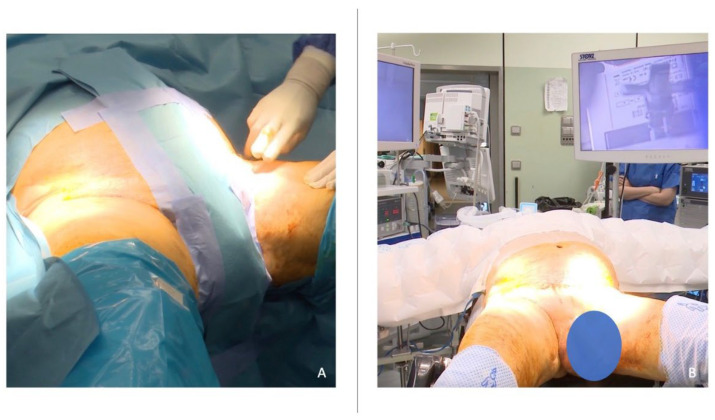
Patient positioning and operating room setting during simultaneous bilateral video–endoscopic lymphadenectomy (sB-VEIL). (**A**) The patient is placed in the supine position with the arms along the body, the hips abducted at 45°, and the knees flexed at 90° using Allen stirrups. (**B**) Two independent consoles and screens are placed contralaterally to the surgical field, and the principal surgeon.

**Figure 2 jcm-12-07272-f002:**
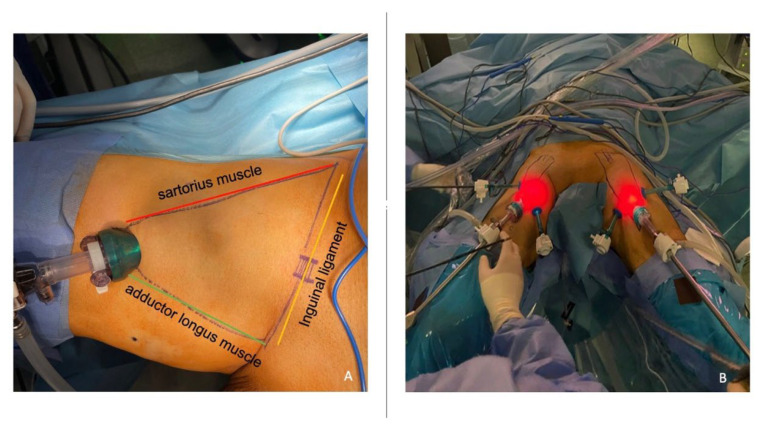
Anatomic surgical landmarks and trocar placement during simultaneous bilateral video–endoscopic lymphadenectomy (sB-VEIL). (**A**) The anatomical landmarks of the inguinal lymph node dissection are the adductor longus muscle (medially), the sartorius muscle (laterally), the Poupart inguinal ligament (superiorly), the Camper’s fascia (superficially), and the femoral vessels and fascia lata (deeply). (**B**) A 3-port configuration is used for both consecutive and simultaneous double video–endoscopic lymphadenectomy. A 12 mm balloon optic trocar is placed approximately 2 cm distal to the apex of the femoral triangle. Two other operative 5 mm balloon trocars are placed approximately 5 cm from the first incision and 2 cm outside the lateral and medial margin of the femoral triangle, respectively.

**Figure 3 jcm-12-07272-f003:**
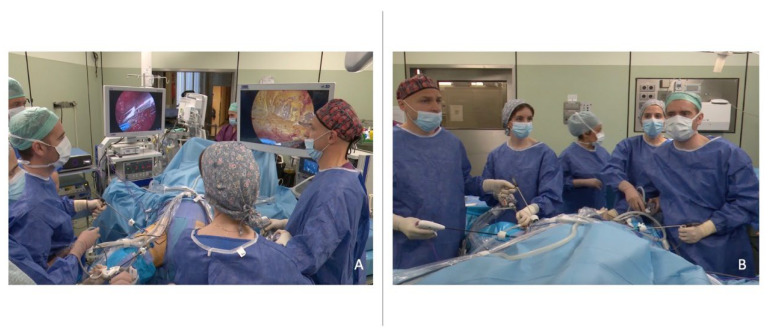
Surgical setting and surgeon placement during simultaneous bilateral video–endoscopic lymphadenectomy (sB-VEIL). Back (**A**) and front (**B**) vision of surgical team performing simultaneous bilateral video–endoscopic lymphadenectomy for penile cancer. The assistants are placed medially on the patient’s leg. The main surgeons are placed laterally on the patient’s leg.

**Table 1 jcm-12-07272-t001:** Baseline and pathological patients’ characteristics according to the surgical approach.

Variable	cB-VEIL (n = 20)	sB-VEIL (n = 10)	*p*-Value
Age, yrs			
Median (IQR)	69 (62.8–76.2)	76.5 (71–79)	0.18
BMI, kg/m			
Median (IQR)	28.1 (27.6–33.7)	26.2 (25–28.5)	0.08
ASA score, n (%)			0.6
1	1 (5)	0 (0)	
2	8 (40)	6 (60)	
3	11 (55)	4 (40)	
Diabetes, n (%)			0.9
No	14 (70)	6 (60)	
Yes	6 (30)	4 (40)	
Smoking status, n (%)			0.8
Never	11 (55)	5 (50)	
Current	5 (25)	3 (30)	
Former	4 (20)	2 (20)	
Primary surgery type, n (%)			0.9
Glandectomy	8 (45)	4 (40)	
Partial penectomy	7 (35)	4 (40)	
Total penectomy	4 (20)	2 (20)	
Primary pT stage, n (%)			0.03
pT1	0 (0)	4 (40)	
pT2	10 (50)	3 (30)	
pT3	10 (50)	3 (30)	
Primary tumor grade, n (%)			0.1
G1	3 (15)	1 (10)	
G2	10 (50)	4 (40)	
G3	7 (35)	5 (50)	
cN stage, n (%)			0.7
cN0	4 (20)	3 (30)	
cN1-2	16 (80)	7 (70)	
LVI, n (%)			0.9
No	16 (80)	7 (70)	
Yes	4 (20)	3 (30)	
pN stage, n (%)			0.6
pN0	14 (70)	6 (60)	
pN1	3 (15)	3 (30)	
pN2	3 (15)	1 (10)	

BMI: body mass index; ASA: American Society of Anesthesiologists; LVI: lymphovascular invasion; cB-VEIL: consecutive bilateral video–endoscopic inguinal lymphadenectomy; sB-VEIL: simultaneous bilateral video–endoscopic inguinal lymphadenectomy.

**Table 2 jcm-12-07272-t002:** Intraoperative and postoperative outcomes according to the surgical approach.

Variable	cB-VEIL (n = 20)	sB-VEIL (n = 10)	*p*-Value
Operative time, min			
Median (IQR)	240 (184–300)	170 (164–180)	<0.01
Total n. LN removed			
Median (IQR)	13.5 (10.8–18)	14 (11.8–15.2)	0.7
Total n. positive LN			
Median (IQR)	0 (0–1.25)	0 (0–1)	0.9
N. LN right			
Median (IQR)	7 (5.75–10.2)	7 (5–8)	0.5
N. positive LN right,			
Median (IQR)	0 (0–1)	0 (0–1)	0.09
N. LN left			
Median (IQR)	7 (5.75–7.5)	7 (6.5–8)	0.7
N. positive LN left,			
Median (IQR)	0 (0–0)	0 (0–1)	0.2
EBL, mL			
Median (IQR)	10 (10–25)	10 (10–25)	0.9
LOS, days			
Median (IQR)	6 (5–7.25)	7 (6.75–8.5)	0.2
Drainage removal, days			
Median (IQR)	28.5 (21.5–50.5)	25.5 (19.2–32.8)	0.2
Wound complication, n (%)			0.7
No	17 (85)	9 (90)	
Yes	3 (15)	1 (10)	
Lymphocele, n (%)			0.9
No	17 (80)	8 (80)	
Yes	4 (20)	2 (20)	
Lymphedema, n (%)			0.9
No	14 (70)	7 (70)	
Yes	6 (30)	3 (30)	
DVT, n (%)			0.8
No	19 (95)	10 (100)	
Yes	1 (5)	0 (0)	
Clavien–Dindo classification, n (%) *			0.6
0	6 (30)	4 (40)	
1	6 (30)	2 (20)	
2	5 (25)	3 (30)	
3a	3 (15)	1 (10)	
3b	0 (0)	0 (0)	
90-d readmission, days			0.9
No	16 (80)	8 (80)	
Yes	4 (20)	2 (20)	

* reported per number of patients included. LN: lymph nodes; LOS: length of hospital stay; EBL: estimated blood loss; cB-VEIL: consecutive bilateral video–endoscopic inguinal lymphadenectomy; sB-VEIL: simultaneous bilateral video–endoscopic inguinal lymphadenectomy.

## Data Availability

Data will be made available on reasonable request by the corresponding author.

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
