# Peer review of "Simultaneous Bilateral Video–Endoscopic Inguinal Lymphadenectomy for Penile Carcinoma: Surgical Setting, Feasibility, Safety, and Preliminary Oncological Outcomes"

_jcm, 2023, doi:10.3390/jcm12237272_

Round 1
Reviewer 1 Report
Comments and Suggestions for Authors
The authors present their experience with a technique already known of simultaneous videoendoscopic inguinal lymphadenectomy. The paper is well-written and easy to follow. The introduction offers a good overview of the subject and the objectives. The methods are well explained, and the results are in accordance with the objectives. The discussion covers important aspects of the research. The only point I raise is that I missed some discussion about the practical aspects of the simultaneous approach, like intercommunicability among the team, ergonomics, and costs (materials and HR). In the study, they used twice as many resources in the simultaneous approach for saving 30% of time, without any extra advantage. What do the authors think about that? Other than that, I congratulate the authors for their efforts in a needy area of uro-oncology.
Author Response
We thank the Reviewer for these important comments. Cost analysis was not an objective of this study, however, it represents an important outcome that should be considered when electing patients for bilateral ILND. Although further data are needed to clearly assess the cost-effectiveness of this procedure, we think that a simultaneous approach could be associated with reduced costs, not only because it is associated with lower operative time, but also because it reduces the anesthesia-related costs and hospitalization costs in patients elected for a bilateral procedure. Furthermore, surgical instruments used are mostly reusable with just single-use sealing devices having an impact on the overall costs.
We have modified the discussion as follows:” although the simultaneous approach requires the use of several surgical instruments twice, most of them are reusable disposable (monopolar scissors, bipolar forceps, cameras) except for single-use sealing devices. Thus, it could reduce surgery-related costs due to the lower operative time and in patients elected for bilateral ILND, it reduced anesthesia-related costs as well as hospitalization costs. Nonetheless, additional data are imperative to thoroughly evaluate the cost-effectiveness of this procedure.”
Regarding the other strengths of this procedure in terms of ergonomics and team work we modify the discussion section as follows:" Furthermore, sB-VEIL surgery may ease the implementation of mentoring programs in laparoscopic surgery since young or inexperienced surgeons could be guided side by side during the surgical steps by experts in the field. In this regard, the surgical setting herein presented allows for an ergonomic procedure in both limbs, without any technical disadvantage or issues compared to cB-VEIL.”

Reviewer 2 Report
Comments and Suggestions for Authors
Though this is not a randomized trial the Authors might expand on the reasons of choosing consecutive versus simultaneous approaches for reader interest
A mention of the Authors' experience regarding cost effectiveness and availability of both equipment and requisite training/expertise would be appreciated
Author Response
We thank the Reviewer for these important comments. We think that the simultaneous approach should be proposed for patients elected to bilateral ILND as it could reduce anesthesia-related complications while being potentially cost-saving. Regarding the experience, no relevant data can be provided since no study has evaluated the learning curve of a laparoscopic ILND. We have modified the discussion as follows: “According to our findings, sB-VEIL is a feasible procedure with a comparable safety profile to cB-VEIL, with the main strength of a reduced operative time and potentially cost-saving. Although the simultaneous approach requires the use of several surgical instruments twice, most of them are reusable disposable (monopolar scissors, bipolar forceps, cameras) except for single-use sealing devices. Thus, it could reduce surgery-related costs due to the lower operative time and in patients elected for bilateral ILND, it reduced anesthesia-related costs as well as hospitalization costs. Nonetheless, additional data are imperative to thoroughly evaluate the cost-effectiveness of this procedure. Furthermore, sB-VEIL surgery may ease the implementation of mentoring programs in laparoscopic surgery since young or inexperienced surgeons could be guided side by side during the surgical steps by experts in the field. In this regard, the surgical setting herein presented allows for an ergonomic procedure in both limbs, without any technical disadvantage or issues compared to cB-VEIL. Finally, although further data are needed to confirm its efficacy in terms of survival, preliminary oncological outcomes are comparable to those of previous series. Thus, taking all these findings together, adopting a simultaneous approach should be considered to mitigate the potential for anesthesia-related complications and to decrease surgical-related expenses. Simultaneous ILND represents an alternative to a consecutive approach, especially in inguinal node-positive patients elected for bilateral ILND with high preoperative anesthesiological risk or in cN0 disease in the absence of non-invasive diagnostic tools such as DSNB.

Reviewer 3 Report
Comments and Suggestions for Authors
TITEL
Is not informative of the study can be improved.
ABSTRACT
Must be shortened.
INTRODUCTION
Well done.
MATERIALS AND METHODS
1. A reference for the American Society of Anesthesiologists (ASA) score used must be added.
2. Methods are described in detail for both Patient positioning, trocar placement, and surgical setting, and Surgical technique and postoperative care better to move the detail to supplementary material and keep only the important techniques.
RESULTS
1. As a total of 30 subjects the statistical analysis showed no significance but still the results are important as they come from the operation room as a new trial however authors can do small sample size adjustment in the analysis so they can have significant results that may influence the results.
DISCUSSION
Clear
CONCLUSION
Supported by results.
Comments on the Quality of English LanguageAcceptable
Author Response
TITLE
Is not informative of the study can be improved.
- We disagree with the Reviewer's comment since we think that the title is comprehensive of all the findings that are reported and discussed within the manuscript
ABSTRACT
Must be shortened.
- We have complied with the editorial requests in terms of abstract length
INTRODUCTION
Well done.
MATERIALS AND METHODS
- A reference for the American Society of Anesthesiologists (ASA) score used must be added.
- We have added the reference as suggested by the Reviewer
- Methods are described in detail for both Patient positioning, trocar placement, and surgical setting, and Surgical technique and postoperative care better to move the detail to supplementary material and keep only the important techniques.
- We thank the Reviewer for this comment. Although surgical technique and postoperative patient care should be moved to a supplementary material section we must comply with the editorial requests of providing a manuscript of at least 4000 words. So, for this reason, we must keep the surgical technique within the main text.
RESULTS
- As a total of 30 subjects the statistical analysis showed no significance but still the results are important as they come from the operation room as a new trial however authors can do small sample size adjustment in the analysis so they can have significant results that may influence the results.
We agree with the Reviewer comment. However, although we relied just on 30 patients for this analysis, penile carcinoma is a rare disease and patients with clinical node-positive disease are even rarer. So, the results of our study are exploratory and further studies will be useful to demonstrate the possible superiority of one of these approaches for ILND. We have acknowledged it in the limitations of the study. Limitations have been modified as follows:” Thus, multicenter studies are needed to demonstrate the feasibility of the technique and the possible advantages of a simultaneous approach in terms of reduced morbidity compared to a consecutive technique, as well as estimate its learning curve in laparo-scopic-naïve or inexperienced surgeons and its cost-effectiveness.”.
DISCUSSION
Clear
CONCLUSION
Supported by results.
